# Extended Trochanteric Osteotomy Does Not Compromise Functional and Radiographic Outcomes of Femoral Stem Revisions with the Use of an Uncemented Modular Conical Stem

**DOI:** 10.3390/jcm13195921

**Published:** 2024-10-04

**Authors:** Tomasz Jopek, Paweł Chodór, Łukasz Łapaj, Waldemar Woźniak, Sławomir Michalak, Jacek Kruczyński

**Affiliations:** Department of General Orthopedics, Muskuloskeletal Oncology and Trauma Surgery, Poznan University of Medical Sciences, ul. 28 Czerwca 1956 r 135/147, 60-545 Poznań, Poland; tomek.satori@wp.pl (T.J.); llapaj@orsk.pl (Ł.Ł.); wwozniak@orsk.pl (W.W.); smichalak@orsk.pl (S.M.); jkruczynski@orsk.pl (J.K.)

**Keywords:** revision hip arthroplasty, extended trochanteric osteotomy, conical stems

## Abstract

**Background**: Stem revisions in revision total hip arthroplasty (THA) with proximal bone stock loss may be dealt with utilizing modular, uncemented conical stems. During stem extraction, surgeons may resort to extended trochanteric osteotomy (ETO). However, ETO is associated with extensive blood loss and infections. This study compared the clinical outcomes, radiographic results and complications in THA revisions utilizing conical modular stem with and without ETO. **Methods:** Patients who underwent revision THA with or without ETO were assessed retrospectively. The minimal follow-up was 3 years. The functional evaluation included Harris Hip Score (HHS) and Short Form 36 (SF-36) as well as Western Ontario and McMaster Universities Arthritis Index (WOMAC) and Numerical Rating Scale for pain assessment. The radiographic evaluation comprised bone defect assessment, osteotomy healing, stem migration and position, presence of radiolucent lines and stress shielding. **Results:** In total, 73 patients (80 hips) were included in the final analysis. The ETO group comprised 48 hips, and the no-ETO group comprised 32 hips. In the ETO group, pre-operative WOMAC scores were lower than in the no-ETO group (*p* = 0.012). No significant differences were found in terms of post-operative HHS, WOMAC, and NRS scores between groups, except worse results were found in the case of claw plate implantation. Patients in the no-ETO group exhibited better results in SF-36 than in the ETO-group. Osteotomy non-union was observed in four hips (9.5%). Stam varus/valgus position was within ±1.5 degrees (85.9%). **Conclusions:** ETO does not adversely impact outcomes in patients undergoing femoral stem revisions with modular conical stems. The invasive nature of these procedures prompts careful consideration in each case individually.

## 1. Introduction

As the number of primary total hip arthroplasties (THAs) is increasing globally, there is a growing demand for revision hip arthroplasties [1,2]. These procedures are technically demanding, time-consuming, and pose a financial burden, particularly in cases involving an exchange of the femoral stem [1,3,4].

Modular uncemented conical stems have emerged as the golden standard in revision THA, especially in cases with proximal bone stock loss [4,5,6]. Most contemporary designs are based on the implant developed by Wagner and feature a fluted conical distal part with a modular proximal component [7,8]. To achieve optimal results, such implants should be fixated distally according to the cone-in-cone principle. The proximal part is then implanted so that leg length and rotation are adjusted independently [4,6,9].

In many situations, performing an extended trochanteric osteotomy (ETO) may facilitate the revision of the femoral stem. This procedure provides controlled access to the femoral canal, with indications including femoral revisions of well-fixed cemented stems or cases involving difficult cement removal. ETO is also beneficial in the revision of well-fixed cementless components, particularly implants with extensive osseointegrative surface finishes [2,4]. Additionally, ETO can facilitate the implantation of modular conical stems in cases of proximal femur deformities and periprosthetic fractures; relative contraindications include impaction grafting and revisions where the stem will be fixed with bone cement [2,4,10,11].

The use of ETO in revision THA may be associated with several potential problems: extended surgical exposure may elevate the risk of bleeding and infection [4,12]. Moreover, the osteotomy reduces the amount of intact bone, potentially compromising the primary stability of the revision stem. While ETOs are frequently utilized during revision THA, there are limited data available comparing revisions with and without ETO using the same type of implant and surgical technique [2,10,13].

This study compared the clinical outcomes, radiographic results, and complications in THA revisions where femoral stems were exchanged for a conical modular stem with and without performing ETO.

## 2. Materials and Methods

### 2.1. Study Design

This retrospective study obtained approval from the local ethics committee (Nr 1181/17) and included patients from our institution who underwent revision total hip arthroplasty (THA) using a modular uncemented conical stem (Stryker Restoration). The patients were categorized into two groups—ETO and no-ETO—depending on whether an extended trochanteric osteotomy was performed. Both groups comprised patients revised for aseptic loosening, periprosthetic infections, periprosthetic fractures, and instability, with a minimum follow-up of 3 years.

This study applied the following exclusion criteria: the absence of pre-operative or post-operative radiographs taken within 2 months before and after the procedure, severe neurological deficits that could impede patients’ participation in the study, and an inability to contact the patient (two mail or e-mail attempts and three phone attempts were made). Additionally, cases where a spacer was implanted during the primary arthroplasty due to intraoperative signs of infection and subsequently revised to a modular stem were also excluded.

### 2.2. Patients

Out of the initially identified 120 patients, successful contact was established with 103 patients or their families. Within this group, 17 patients had passed away (none of the deaths were directly related to the revision procedures); 4 patients declined to participate in the study, or their neurological status did not allow for participation; in 5 cases, the stems were revised due to infections and implant fracture; and in 4 cases, patients were excluded due to incomplete pre-operative radiographic data. Consequently, 73 patients (80 hips) were included, and their demographics are presented in Table 1. Among the 7 patients who underwent bilateral revisions, in 5 cases, an ETO was performed in one hip, in 1 patient, a bilateral ETO was performed, and 1 patient had both stems revised without osteotomies.

### 2.3. Revision Procedures

Patients included in this study underwent stem as well as stem and cup revisions; various bone stock defects were present in this group (Table 2). The following revision causes were identified with data for ETO/no-ETO groups provided in brackets: aseptic loosening—47 cases (23/24); periprosthetic fractures—18 cases (7/11); periprosthetic infections—6 cases; two-stage revisions (2/4); fractures of femoral stems—4 cases (4/0); recurrent dislocations—2 cases (2/0); and leg length discrepancy—2 cases (1/1).

### 2.4. Surgical Technique

The revisions were performed by five experienced surgeons using a posterior approach, all of whom were familiar with ETO techniques. In all cases, a consistent and algorithmic surgical technique was employed. If digital pre-operative planning (Orthoview software, Materialise, Leuven, Belgium) demonstrated severe proximal femur deformities, such as varus deformation related to loosening and subsequent bone remodelling, an ETO was performed electively to facilitate femoral reaming. Otherwise, the procedure began with an attempt at endofemoral removal, with an ETO conducted if the initial approach was unsuccessful.

The technique varied for cemented and cementless stems. In all cases, the pseudomembrane tissues surrounding the stem were resected to expose the entrance to the femoral neck. The stem was then firmly grasped using heavy-duty Vise-Grips, and the surgeon evaluated whether it was well fixed or loose, based on pre-operative radiographs.

For loose stems, the femoral neck was cleared of soft tissues, and a narrow chisel was used to release the stem, typically allowing for its removal. If bone cement remained attached, its fragments were extracted using a specialized instrument set (Stryker Gray Revision Set), which included retrograde hooks and reamers. An arthroscope was employed when necessary to ensure complete cement removal.

For well-fixed cemented stems with a smooth finish, the stem was removed with several gentle taps, and the cement mantle was overdrilled and extracted as described above. However, cemented stems with a rough surface were more challenging to remove, and usually after several unsuccessful taps, an ETO was performed.

During revisions of well-fixed cementless stems, the proximal part of the stem was first released from the bone using thin, narrow chisels, except in cases where the stem had extensive collars. For extensively coated stems and stems with large collars, thin K-wires were introduced along the stem in multiple areas to assist in its release from the bone. As most contemporary stems feature a threaded hole for implantation or removal, a universal slap hammer with threaded attachments or grips for holding the femoral neck (ChM, Poland) was used to extract the stem. If multiple attempts were unsuccessful, an ETO was performed.

In 48 cases where ETOs were carried out, a consistent surgical technique was used; bony cuts were carried out on the posterolateral part of the femur, following the technique described by Paprosky [4]. In brief, a longitudinal incision was made on the posterior side of the femur, extending to the end of the revised stem or bony deformity. Subsequently, a transverse incision was made on the lateral cortex. Then, by inserting several chisels, a flap of the lateral cortex was raised exposing the femoral canal. After the stem’s removal, typically, a “prophylactic loop” of cable (Stryker Dall-Miles system) was placed approximately 1 cm below the transverse incision to mitigate the risk of cortical cracking during subsequent steps. Then, the distal bone was reamed to allow for distal stem insertion; trial implants were then used to adjust leg length and femoral orientation. Once the final implant was in situ, bony fragments were stabilized using cables. In six hips, additional fixation using a claw plate was required as the fixation of the greater trochanter was considered unsatisfactory. The decision on whether to perform the osteotomy was based on the surgeons’ own preference and intraoperative evaluation of stem stability.

### 2.5. Outcome Measures

All patients were invited to routine follow-up visits at two weeks (wound inspections) and two months (radiographic and clinical evaluation) and were advised to conduct subsequent evaluations every 1–2 years. For this study, an additional visit was scheduled and included radiographic and functional evaluations. The functional evaluation included the Harris Hip Score (HHS: 0—worst; 100—best outcome) and a quality of life (QOL) assessment using Short Form 36 (SF-36). Since the SF-36 questionnaire focuses on general health, all analyses were performed concerning 73 patients who were divided into those with no ETO or ETO in at least one hip. The results of all domains were translated into percentages (0%—worst; 100%—best QOL). Additionally, the Western Ontario and McMaster Universities Arthritis Index (WOMAC score: 0—worst; 96—best outcome) and pain evaluation using the Numerical Rating Scale (NRS score: 0—no pain; 10—maximal pain) scores were obtained pre-operatively and at the follow-up visit.

Radiographic evaluation was based on conventional anteroposterior and lateral radiographs performed pre-operatively, post-operatively, at 2 months post-operatively, and during the follow-up visit. All DICOM images were initially calibrated using the femoral head as a reference. Qualitative and semi-quantitative parameters, such as osteotomy healing or bone defect classification according to Paprosky, were independently evaluated by two experienced orthopedic surgeons (ŁŁ and TJ) [14]. In cases where differences in evaluations were present, a final decision was reached through consensus among researchers. The extent of bony defects was assessed based on pre-operative radiographs, with post-operative images serving as the reference for the post-operative evaluation. This assessment encompassed measurements of the following parameters: (1) migration of the stem, (2) radiographic union of the osteotomy, (3) deviation of stem orientation with respect to the axis of the femoral shaft, (4) presence of radiolucent lines, bone resorption, and cortical thickening, and (5) stress shielding.

### 2.6. Statistical Analysis

Statistical analysis was performed using the SPSS package version 26 (IL, USA) and GraphPad Prism version 10.1 software (MA, USA), with a significance level set at *p* < 0.05. The normality of distribution was confirmed using the Shapiro–Wilk test; if confirmed, differences between groups were assessed using the independent t-test. Otherwise, the Mann–Whitney U test and Kruskall–Wallis test with post hoc Dunn evaluation were applied. Within-group comparisons were conducted using the Wilcoxon test, and contingency analyses were performed using Fisher’s exact test.

## 3. Results

### 3.1. Demographic Data and Pre-Operative Functional Evaluation

Statistical analysis revealed no significant differences between the demographic parameters of both groups, including age, BMI, and follow-up time (Mann–Whitney test). Similarly, the distribution of genders, patients aged less than 75, those aged 75 or more years, types of revisions, and the incidence of femoral defects did not differ between the two groups (Fisher’s exact test).

During the pre-operative evaluation, WOMAC scores were significantly lower than the values obtained at follow-up (Wilcoxon test, Figure 1a), while NRS pain scores were significantly higher than those obtained at follow-up (Wilcoxon test, Figure 1b). In patients who later underwent ETO, WOMAC scores were significantly lower than in patients without osteotomy (Mann–Whitney test, Figure 1c); however, there were no differences in NRS pain scores between both groups (Mann–Whitney test, Figure 1d).

### 3.2. Functional Assessment at Follow-Up

In this study, no significant differences were observed in HHS results between patients who underwent ETO (mean score 73.9) and those without (mean score 65.5) osteotomies (determined by the Mann–Whitney test, Figure 2a). Within the entire cohort, there were no variations in HHS results between patients aged less than 75 years and those aged 75 years or older (Student’s *t*-test). Additionally, no significant differences were found between patients undergoing stem or cup and stem revisions, as well as between revisions of cemented and cementless stems (Mann–Whitney test). However, the use of a claw plate was associated with lower HHS scores compared to other patients (Mann–Whitney test; *p* < 0.05).

WOMAC scores were comparable between patients who underwent ETO (mean score 52.8) and those without (mean score 60.2) osteotomy (Mann–Whitney test, Figure 2b). Similarly, in the entire cohort, no differences were observed between patients younger than 75 years and those aged 75 years or more (Student’s *t*-test). Additionally, there were no significant differences between patients undergoing revision of cup and stem versus stem only, as well as between revisions of cemented and cementless stems (Mann–Whitney test). However, cases involving the implantation of a claw plate showed significantly lower WOMAC scores (Mann–Whitney test, *p* < 0.05).

No significant differences were found in NRS scores between patients without osteotomies and those who underwent ETO (Mann–Whitney test, Figure 2c). In the entire cohort, patients aged 75 years or more exhibited pain levels comparable to their younger counterparts (Student’s *t*-test). Furthermore, revisions involving stems or a cup and stem, revisions of cemented or cementless stems, and the use of a claw plate did not significantly affect pain levels (Mann–Whitney test).

### 3.3. Quality of Life

There were differences between certain aspects of HR-QOL in patients who underwent revisions with and without ETO (Figure 3). In the latter group, scores were significantly higher for general health and physical functioning (both Student’s t-tests, *p* < 0.05) as well as for the role limitation emotional domain (Student’s t-test, *p* < 0.01).

Post-operative function was evaluated using the HHS and WOMAC questionnaires correlated with SF-36 domains except for HHS scores in patients who did not undergo ETO. Correlation coefficients for physical and mental scores were moderate in all cases (Table 3).

The use of a claw plate negatively affected the physical functioning domain (Mann–Whitney test, *p* < 0.05) but had no impact on other domains. In patients where both the stem and acetabular components were exchanged, the vitality domain had significantly lower scores (Student’s t-test, *p* < 0.05), while other domains remained unaffected. There were no differences in the QOL of patients younger than 75 and those aged 75 or older, as well as patients with revisions of cemented or uncemented stems (Student’s t-test in both cases).

### 3.4. Radiographic Evaluation

In the majority of patients, the implantation of the stem resulted in good primary stability, with stem migration observed in only two cases. Migration was determined based on two reference points. The tip of the greater trochanter served as the primary reference point for each patient. In cases where no ETO was performed, the tip of the lesser trochanter was also used. In hips where ETO was performed, characteristic locations of cables were chosen as reference points. The results were averaged in each case.

In one patient, stem subsidence of 14 mm was detected on a radiograph taken at 8 weeks post-op. Since the hip was asymptomatic, the decision was made not to revise the implant, and at follow-up (5 years, 2 months), no subsequent migration was observed, and the patient exhibited good hip function. In a different patient, a 2 mm migration was observed at 8 weeks, with no subsequent migration noted at the 6-year follow-up.

Radiographic union was defined by Abdel et al. as the presence of a callus bridging the osteotomy or the disappearance of the osteotomy line on orthogonal radiographs (Figure 4) [4].

In this study, during the final evaluation, a lack of osteotomy union on any of the orthogonal radiographs was observed in four hips (9.5%). However, this phenomenon was not associated with decreased functional outcomes, and none of these patients required revision procedures (Figure 5a–c). In five hips, despite careful surgical techniques, large cortical bone fragments denuded of muscle were inadvertently broken off intraoperatively. Each of these devascularized fragments was subsequently attached to the femoral stem, and in all cases, osseointegration was observed (Figure 5d–f).

In the entire cohort, stem orientation was evaluated as deviation between the axis of the stem and the axis of the femoral canal measured in the central part of the stem on AP radiographs. At follow-up, it ranged between 3 degrees valgus and 7 degrees varus, while in 67 cases (85.9%), stem orientation evaluated at follow-up was within the range of ±1.5 degrees (Figure 4). In one case, implantation of the stem in relative varus caused penetration of the lateral cortex; however, this was not associated with stem migration at follow-up, and there were no differences between stem orientation in patients with and without ETO (Mann–Whitney test)

The presence of cortical thickening (1 mm or more), bone resorption corresponding to stress shielding, and radiolucent lines (0.5 mm or more) was examined on orthogonal radiographs taken at follow-up visits to determine bone remodelling around the stem. All features were examined in seven Gruen zones with divisions between zones set at the junction of proximal and distal components and in the middle of the distal component. Bone radiolucencies and resorption were observed predominantly in proximal zones, while cortical thickening was observed predominantly in distal areas (Figure 6). In both groups, there was no progression of bony defects between post-operative and follow-up radiographs.

Stress shielding was quantified according to criteria developed by Engh and modified by Huang et al. [10]. Briefly, depending on the extent of bone resorption (number of zones where it was present), grades one to four were assigned. There was no difference in the incidence of stress shielding between hips with and without ETO (Mann–Whitney test; see Figure 7). Likewise, there were no differences in stress shielding between males and females, patients aged less than 75 or 75 or more years old, or cases involving the revision of cemented and uncemented stems (Mann–Whitney tests in all cases). The extent of stress shielding did not correlate with the diameter of the stem (Spearman test). In cases where ETO was performed, there was no correlation between stress shielding and NRS pain scores (Spearman test). However, in cases where ETO was not performed, stress shielding correlated with NRS and HHS scores (Spearman test; *p* = 0.0143, r = 0.4290; *p* = 0.0008, r = −0.5608, respectively).

### 3.5. Complications

Several types of complications were observed among the 103 patients included in this study. In four cases (3.9%), stems were revised due to periprosthetic infections; one patient underwent a two-stage revision with another modular stem, two patients opted for implant removal only, and one patient underwent the first stage of a two-stage revision at follow-up. Notably, none of the aforementioned patients were included in the study cohort.

Additionally, one patient experienced two dislocations within the first 6 weeks post-op, and another patient required the exchange of the acetabular liner to a constrained component due to multiple dislocations. Intraoperative fractures occurred in eight cases, and as mentioned previously, in four hips, a devascularized fragment was broken off. These fragments were fixed using Dall-Miles cables; however, four of these patients required the use of a claw plate. Another patient underwent a periprosthetic fracture below the stem after falling down the stairs 8 weeks following surgery. The fracture was successfully managed using a dedicated locking plate (Zimmer NCB plate), resulting in a good union.

## 4. Discussion

Extended femoral osteotomies are very useful in cases where the extraction of a well-fixed stem poses a technical challenge [2,4,10]. ETOs can also assist in the removal of bone cement fragments and facilitate the implantation of revision stems in situations involving severe proximal femur deformities [4,5]. However, as demonstrated by previous studies, a considerable number of stem revisions can be effectively executed without the necessity of an osteotomy, potentially mitigating the invasiveness of the procedure [5,7,10,15].

This study conducted a comparative analysis of two patient groups who underwent revision procedures employing the same implant and were conducted by the same surgical team, either with or without ETO. The findings indicate that performing the osteotomy does not have a discernible impact on the clinical and radiographic outcomes of the revision procedure.

There are several limitations to this study. Firstly, it is a retrospective analysis, and the decision on whether a particular ETO was performed in a given patient was susceptible to bias. Nevertheless, it should be noted that both groups exhibited comparable demographics, incidence of bony defects, and revision types. It should also be noted, that some patients were lost to follow-up, and it could be expected that their functional outcome was poorer than that of the remaining group. Another limitation is associated with the fact that data regarding the use of the claw plate come from a small number of hips, primarily in cases where the greater trochanter could not be securely fixed using cables. Consequently, poorer outcomes could be expected, as this technique was utilized in technically challenging cases. A further limitation is linked to the fact that, while WOMAC scores were obtained pre- and post-operatively on a routine basis, HHS evaluation was only conducted post-operatively, as required by local healthcare financing regulations. Additionally, radiographic evaluation based on orthogonal radiographs has limited precision, particularly when using semi-quantitative scoring systems; however, this limitation applies to most papers in this field [4,10].

The functional outcomes of patients evaluated in this study using the HHS and WOMAC scores did not exhibit significant differences between those with and without ETO [1,4,11,13,16,17,18]. However, lower scores were observed in patients who underwent the osteotomy. The mean scores were comparable to those reported in other studies employing ETOs, as well as in studies where fluted conical stems were implanted without performing ETO, and in cohorts where ETO was used in a small percentage of patients.

Interestingly, a recent paper by Abdel et al. suggested that in cases where ETO is performed, the surgical technique (either according to Wagner or Paprosky) does not influence clinical outcomes. Several authors have demonstrated that factors such as age and gender may influence the outcomes of primary and revision arthroplasties; however, this was not confirmed in our study, as age, gender, and revision type did not affect the functional outcomes of the revisions [11,13,17].

NRS pain scores in this study were relatively high compared to the pain domain scores of WOMAC, SF-36, and HHS. Interestingly, in other studies on femoral stem revisions using modular stems, comparable outcomes of these questionnaires were associated with lower Visual Analog Scale (VAS) pain scores [18,19,20]. Similar discrepancies between pain levels measured using VAS scores and questionnaires evaluating functional outcomes were also reported for neck pain [21].

In this study, ETOs had a limited negative effect on certain domains of the SF-36 questionnaire; however, values for both groups of patients were comparable to those presented by other authors [16,17]. There was a moderate correlation between post-operative functional scores measured using HHS and WOMAC questionnaires, and similar results were presented by Zampelis et al. for revision total hip arthroplasty (THA) using a modular stem [18].

In the current study, radiographic outcomes were satisfactory in both groups: stem migration occurred in two cases, and most ETOs healed with a union. Similar or slightly higher union rates were reported in other studies; however, migration rates reported in these series varied, with some papers demonstrating higher values [2,4,10,11].

Regardless of whether an ETO was performed, bone remodelling around the stems followed a similar pattern; typically, there was a distal fixation with cortical hypertrophy and bone resorption around the proximal component, with few radiolucencies observed at follow-up [4,10,22,23]. Similar patterns have been described for other types of modular conical stems, both with and without the use of ETO. Some authors reported a high incidence of spot welds near the tips of revision stems and at the junctions of the proximal and distal components [6,10]. This feature was not common in the present study, and this may be explained by a different implant geometry (longer proximal part and a partially conical distal component) in the cited publications. The authors suggested that this did not allow for a very tight press fit and consequently resulted in the formation of spot welds [6,10].

The incidence of complications related to revision total hip arthroplasties (THAs) using conical modular stems varies significantly across the literature, with the most common complications including infections, dislocations, and intraoperative fractures, similarly as in this cohort [4,5,10,11,24]. The current study may suggest that the use of a claw plate may be associated with poorer clinical and radiographic outcomes. However, as this technique was used in challenging cases, the conclusions cannot be generalized to a broader population; moreover, in studies where such plates were used routinely, there was no noticeable negative effect on clinical outcomes [2,3,25].

In this study, modular stems were used in all hip revisions. However, multiple previous reports have indicated good clinical outcomes with the use of monoblock stems [20]. The use of modular stems is potentially associated with several complications—most notably breakage and corrosion at the tapered junctions [20,26]. Modular stems consist of proximal components with various diameters, allowing for more physiological loading of the proximal femur. However, there is strong evidence suggesting that this advantage may be theoretical [9,20].

Several authors have demonstrated that, in comparison to modular stems, monoblock components are associated with better osseous regeneration [10,27,28]. Huang et al. explored this by comparing a series of revisions using monoblock and modular stems. In patients with monoblock stems, relatively long and thin components were used, often achieving three-point fixation. In contrast, the use of modular stems resulted in the implantation of shorter, more rigid components with a larger distal diameter, which were fixed using the “cone-in-cone” principle. This led to distal loading of the femur and increased stress shielding.

There are, however, significant advantages to modular components. First, the ability to adjust the proximal part of the stem compensates for discrepancies in the depth of implantation, which may vary from the pre-operative plan and intraoperative evaluation using trial components. Modularity also allows for precise intraoperative adjustment of component anteversion and offset. Additionally, in cases of post-operative instability, revisions are often simpler, as only the proximal part of the stem may need to be exchanged.

## 5. Conclusions

This study provides evidence that the utilization of ETO does not adversely impact functional and radiographic outcomes, or HR-QOL in patients undergoing femoral stem revisions with modular conical stems. Consequently, if an endofemoral stem removal cannot be easily performed, surgeons should not hesitate to make a decision to conduct ETO as it is likely not going to compromise the clinical and radiographic outcomes of the revision. While the findings affirm the use of ETO in these procedures, its invasive nature prompts careful consideration in each case individually.

## Figures and Tables

**Figure 1 jcm-13-05921-f001:**
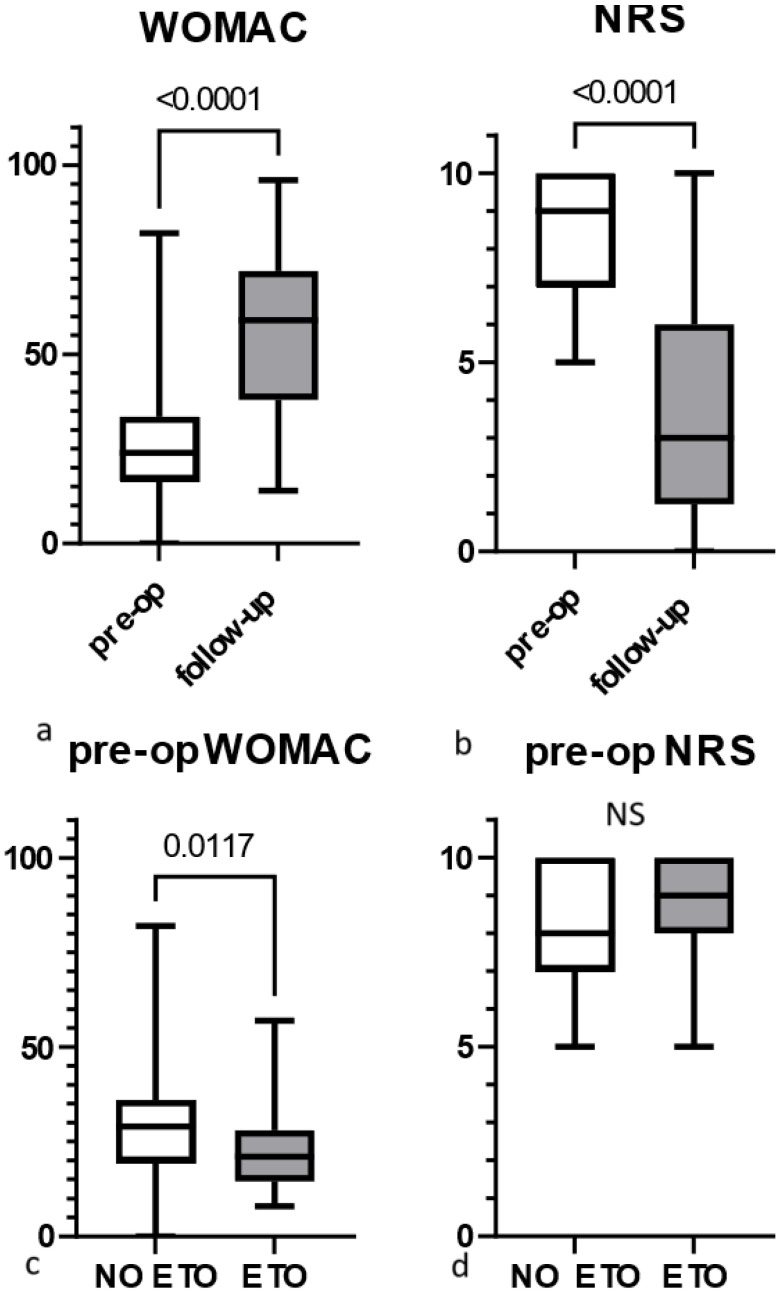
Pre-operative functional evaluation of patients: (**a**) WOMAC scores obtained pre-operatively and at follow-up; (**b**) NRS scores obtained pre-operatively and at follow-up; (**c**) pre-operative WOMAC scores in hips with and without ETO; and (**d**) pre-operative NRS scores in patients with and without ETO. Brackets above the plot indicate the results of statistical analysis: *p*-values are presented in cases where a significant difference was found; “NS” denotes no significant difference. Boxes extend between the 25th and 75th percentiles, while whiskers correspond to maximum/minimum values; the transverse line in the box represents the median value.

**Figure 2 jcm-13-05921-f002:**
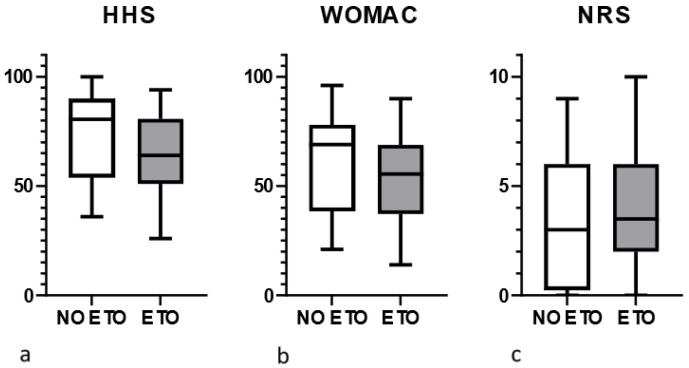
Functional evaluation at follow-up: (**a**) HHS scores in cases with and without ETO; (**b**) WOMAC scores in hips with and without ETO; and (**c**) NRS scores in patients with and without ETO. Boxes extend between the 25th and 75th percentiles, while whiskers correspond to maximum/minimum values; the transverse line in the box represents the median value.

**Figure 3 jcm-13-05921-f003:**
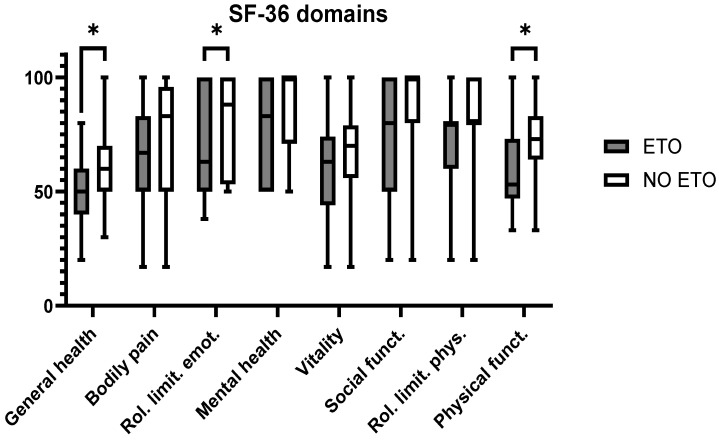
Domains of the SF-36 questionnaire; asterisks indicate statistically significant differences between domains.

**Figure 4 jcm-13-05921-f004:**
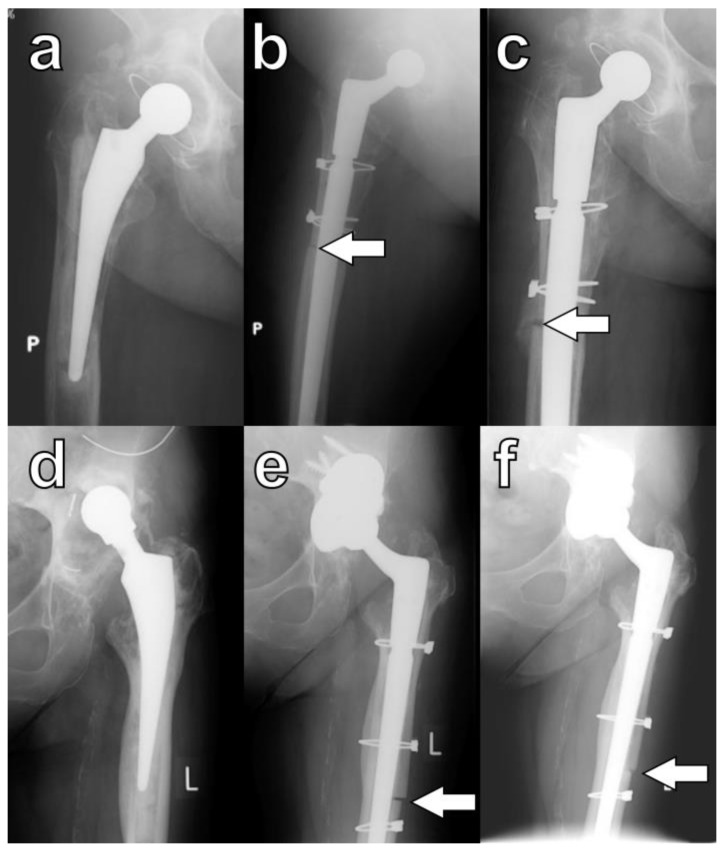
Healing of osteotomies as defined by Abdel et al. [4]: callus formation in serial radiographs of a patient K.W. taken (**a**) pre-op, (**b**) at six months, and (**c**) at eleven months; arrows indicate the formation of a callus at the osteotomy site. Bridging of the osteotomy site: serial radiographs of patient B.K. taken (**d**) pre-op, (**e**) at three months, and (**f**) at twelve months; arrows indicate direct bridging of the osteotomy site. P is an indicator for right side and L is an indicator for left side.

**Figure 5 jcm-13-05921-f005:**
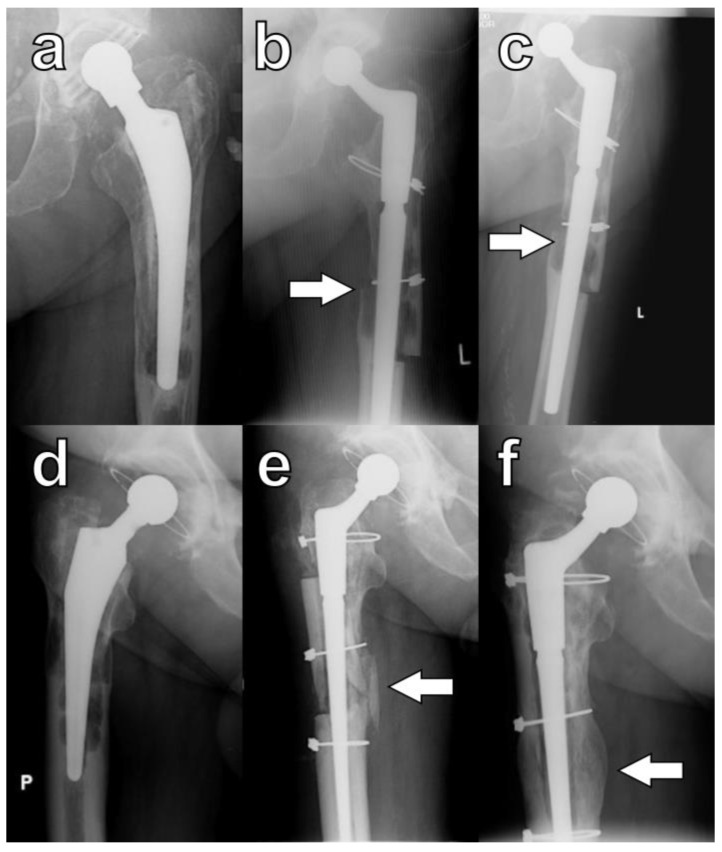
Specific cases of ETO healing: lack of bone formation within the osteotomy (**a**) pre-op, (**b**) at six months, and (**c**) at twenty-four months. Osseointegration of a devascularized bone fragment: (**d**) pre-op, (**e**) at three months, and (**f**) at six months; arrows indicate osteotomy sites. P is right side, and L is left side.

**Figure 6 jcm-13-05921-f006:**
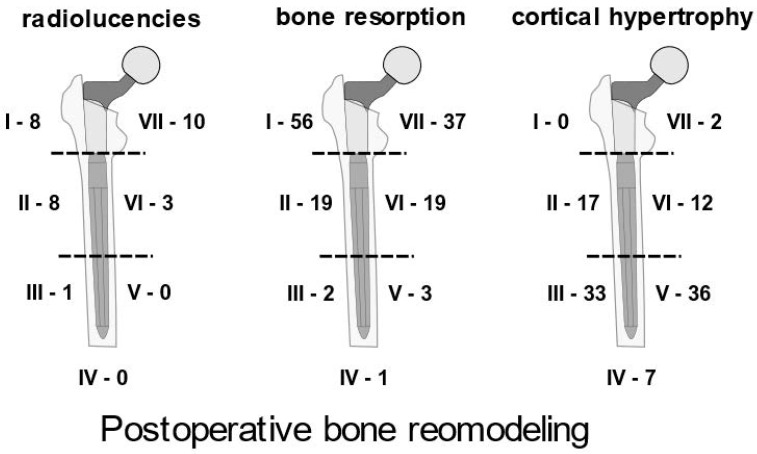
Schematic diagram illustrating the incidence of radiolucencies, bone resorption, and cortical hypertrophy in all hips from this study. Roman numerals denote Gruen zone number, while Arabic numerals represent the number of hips where a specific feature was observed, either on AP or lateral radiographs.

**Figure 7 jcm-13-05921-f007:**
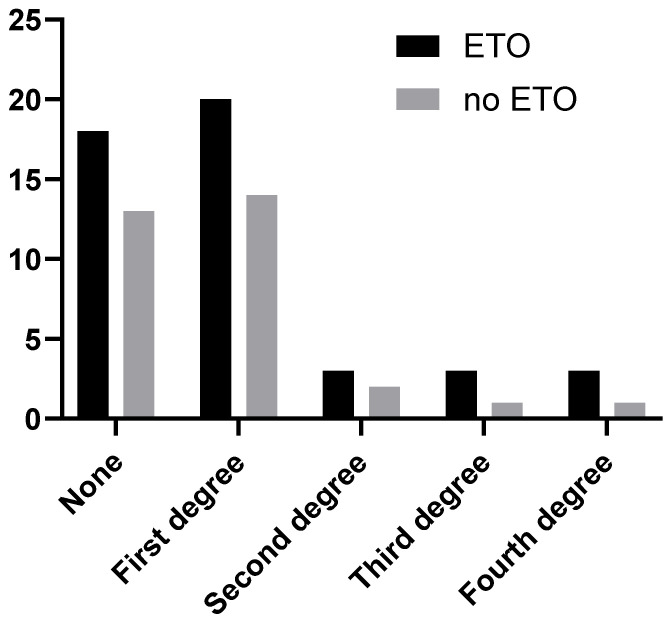
Incidence of stress shielding in patients with and without ETO.

**Table 1 jcm-13-05921-t001:** Comparison of demographic parameters in both groups of patients. An asterisk indicates values given as means with ranges provided in parentheses.

Parameter	ETO Group (48 Hips)	no-ETO Group (32 Hips)
Sex: males/females	32/16	19/13
Mean age at surgery *	72.5 (42–86)	73.3 (44–90)
Patients aged <75/75+	23/25	16/16
Mean BMI at surgery *	27.6 (23.4–33.2)	27.2 (23.3–34.6)
Mean follow-up (in years) *	6.1 (3.0–10.2)	5.8 (3.0–8.3)

**Table 2 jcm-13-05921-t002:** Comparison of parameters describing the revision procedures in both groups of patients.

Parameter	ETO Group (48 Hips)	no-ETO Group (32 Hips)
Stem only/cup and stem revision	19/29	16/16
Revision of cemented/cementless stem	19/29	13/19
Femoral defects (Paprosky) I/II/IIIA/IIIB/IV	7/14/13/11/3	4/9/13/4/2
Hips with a claw plate	4	2

**Table 3 jcm-13-05921-t003:** Correlation coefficients of functional scores (HHS and WOMAC) and SF-36 domains in both patient groups. The upper line of each cell contains the correlation coefficients (r values), while *p* values are provided in brackets. The Spearman test was primarily used for analyses; cases where the Pearson test was employed are denoted by an asterisk.

Correlation	General Health	Bodily Pain	Role Limit. Emotional	Mental Health	Vitality	Social Functioning	Role Limit. Physical	Physical Functioning
no-ETO HHS vs. SF-36	0.2656 (0.1487)	0.0092 (0.9612)	0.1468 (0.4228)	0.285 (0.1138)	0.2806 (0.1198)	0.05328 (0.7721)	0.04895 (0.7902)	0.05615 (0.7602)
no-ETO WOMAC vs. SF-36	0.3861 (0.0290)	0.5492 (0.0011)	0.6733 (<0.0001)	0.6409 (<0.0001)	0.4469 (0.0091)	0.6280 (<0.0001)	0.5106 (0.0024)	0.5120 (0.0023)
ETO HHS vs. SF-36	0.3576 (0.0126)	0.5874 (<0.0001)	0.5648 (<0.0001)	0.5504 (<0.0001)	*0.5139 *(0.0002)	0.6377 (<0.0001)	0.5694 (<0.0001)	* 0.6004 * (<0.0001)
ETO WOMAC vs. SF-36	0.2986 (0.0392)	0.4952 (0.0003)	0.6260 (<0.0001)	0.6545 (<0.0001)	*0.6215 *(<0.0001)	0.6687 (<0.0001)	0.4384 (<0.0001)	* 0.7468 * (<0.0001)

## Data Availability

Data used in this study are available upon request—pchodor@orsk.pl.

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
