# Peer review of "Extended Trochanteric Osteotomy Does Not Compromise Functional and Radiographic Outcomes of Femoral Stem Revisions with the Use of an Uncemented Modular Conical Stem"

_jcm, 2024, doi:10.3390/jcm13195921_

Round 1

Reviewer 1 Report

Comments and Suggestions for Authors

- authors should consider expanding the introduction section with details related to clinical decision making process regarding when to perform ETO in revision THA

- can you detail more on how does the use of modular uncemented conical stems influence the outcomes of revisions with and without ETO?

- what is the anticipated clinical impact of your study findings on clinical practices?

- please define the criteria that surgeons used to decide whether to perform an ETO during revision THA or not?

- describe the surgical technique used for the no-ETO group

- did the authors implement any standardized protocols for followup visits?

- are the authors considering any biases related to lost patients?

- results are close to perfectly presented

- references are up to date

- title seems to be relevant

Author Response

RESPONSE TO REVIEWERS

We would like to thank the referees for their kind reviews and suggestions as they improved the quality of the text.  We implemented nearly all of the suggestions raised in the review as described below :

Reviewer 1.

  1. Authors should consider expanding the introduction.

We included a brief description of the revision protocol in the introduction – the ETO was performed after several attempts to perform an endofemoral revision; an algorithmic approach was used which is described in the materials and methods – surgical technique section

  1. Can you detail more on how does the use of modular uncemented conical stems influence the outcomes of revisions with and without ETO?

The reviewer raised an important point - both modular and monoblock stems have their specific advantages and disadvantages, predominantly related to bone remodelling. We discussed this in the final part of the discussion.

  1. What is the anticipated clinical impact of your study findings on clinical practices?

We believe that this study should encourage other surgeons to perform ETO whenever endofemoral revision is not possible or would be difficult – we modified the conclusions accordingly.

  1. Please define the criteria that surgeons used to decide whether to perform an ETO during revision THA or not?

They were described in the methods section – as mentioned in (1)

  1. Describe the surgical technique used for the no-ETO group

Surgical technique used for non-ETO group was described in Surgical technique subsection (Materials and methods).

  1. Did the authors implement any standardized protocols for follow-up visits?

Yes we did – the patients were seen at 2 weeks and 2 months by the surgeon, then they were advised to schedule visits every 1..2 years. The latest follow-up visit was scheduled for the study.  We described this Outcome measures section

  1. Are the authors considering any biases related to lost patients?

That is a valid point. Most of patients lost to follow-up died from causes not related to the revision. Still it is likely that their functional outcomes and QOL could have been poorer in comparison to the rest of the group. We briefly discussed this in the limitations section.

  1. Results are close to perfectly presented.

 Thank you for the kind comment – we still improved some parts as indicated by Reviewer 2

  1. References are up to date – thank you for the kind comment.
  2. Title seems to be relevant – thank you for the kind comment.

Reviewer 2 Report

Comments and Suggestions for Authors

Thank you for the opportunity to review the article entitled "Extended Trochanteric Osteotomy Does Not Compromise Functional And Radiographic Outcomes Of Femoral Stem Revisions With The Use Of An Uncemented Modular Conical Stem".

The abstract and the "Introduction" section briefly but effectively introduce the reader to the content discussed in the rest of the article - I have no objections to these sections.

The "Materials and methods" section outlines a properly designed study, including appropriate methodologies for statistical analysis.

In the "Results" section, the following elements raise reservations:

- the authors should exclude the superficial information labeled as "not significant" - in Figure 1, the authors should provide the p value for pre-op NRS, and in Table 3, in the row "no-ETO HHS vs. SF-36", the authors should provide both the correlation coefficient and the p value.

- the visual presentation of Figure 2 lacks aesthetic preparation, necessitating modifications to the formatting and internal arrangement of the graphs.

The "Discussion" and "Conclusions" sections do not raise any objections, they fulfill their proper roles.

The work makes positive impressions, but in my opinion it requires a major revision before it can be considered for publication. I will gladly undertake second review after appropriate corrections. 

I congratulate the authors on their work and wish them success in their further scientific careers.

Author Response

RESPONSE TO REVIEWERS

We would like to thank the referees for their kind reviews and suggestions as they improved the quality of the text.  We implemented nearly all of the suggestions raised in the review as described below :

Reviewer 2.

  1. The abstract and the "Introduction" section briefly but effectively introduce the reader to the content discussed in the rest of the article - I have no objections to these sections.

Thank you for the kind comment

  1. The "Materials and methods" section outlines a properly designed study, including appropriate methodologies for statistical analysis.

Thank you for the kind comment

  1. In the "Results" section, the following elements raise reservations:

  1. The authors should exclude the superficial information labeled as "not significant" - in Figure 1, the authors should provide the p value for pre-op NRS, and in Table 3, in the row "no-ETO HHS vs. SF-36", the authors should provide both the correlation coefficient and the p value.

We modified figure 1 and table 3 accordingly

  1. The visual presentation of Figure 2 lacks aesthetic preparation, necessitating modifications to the formatting and internal arrangement of the graphs.

This problem was related to formatting of the image. This was corrected; we also slightly modified markings on images with the radiographs (arrows and letters with improved contrast)

  1. The "Discussion" and "Conclusions" sections do not raise any objections, they fulfill their proper roles.

Thank you, we added some points regarding limitations and the use of modular stems as suggested by reviewer 1

  1. The work makes positive impressions, but in my opinion it requires a major revision before it can be considered for publication. I will gladly undertake second review after appropriate corrections. 

Thank you for the kind comment.

  1. I congratulate the authors on their work and wish them success in their further scientific careers.

The authors are grateful for the kind comments .

Round 2

Reviewer 2 Report

Comments and Suggestions for Authors

Thank you for the opportunity to second review the manuscript.

The authors satisfactorily addressed my prior comments - I have no further objections.

I recommend that the paper be accepted for publication.